# Contribution of Symptomatic, Herbal Treatment Options to Antibiotic Stewardship and Microbiotic Health

**DOI:** 10.3390/antibiotics11101331

**Published:** 2022-09-29

**Authors:** Bernhard Nausch, Claudia B. Bittner, Martina Höller, Dimitri Abramov-Sommariva, Andreas Hiergeist, André Gessner

**Affiliations:** 1Bionorica SE, Research and Development, Kerschensteinerstraße 11-15, 92318 Neumarkt in der Oberpfalz, Germany; 2Institute of Clinical Microbiology and Hygiene, University Hospital Regensburg, Franz-Josef-Strauss-Allee 11, 93053 Regensburg, Germany

**Keywords:** herbal drugs, gut microbiota, antibiotic stewardship, uncomplicated infection, NSAID, homeostasis

## Abstract

Epithelial surfaces in humans are home to symbiotic microbes (i.e., microbiota) that influence the defensive function against pathogens, depending on the health of the microbiota. Healthy microbiota contribute to the well-being of their host, in general (e.g., via the gut–brain axis), and their respective anatomical site, in particular (e.g., oral, urogenital, skin, or respiratory microbiota). Despite efforts towards a more responsible use of antibiotics, they are often prescribed for uncomplicated, self-limiting infections and can have a substantial negative impact on the gut microbiota. Treatment alternatives, such as non-steroidal anti-inflammatory drugs, may also influence the microbiota; thus, they can have lasting adverse effects. Herbal drugs offer a generally safe treatment option for uncomplicated infections of the urinary or respiratory tract. Additionally, their microbiota preserving properties allow for a more appropriate therapy of uncomplicated infections, without contributing to an increase in antibiotic resistance or disturbing the gut microbiota. Here, herbal treatments may be a more appropriate therapy, with a generally favorable safety profile.

## 1. Introduction

Epithelial surfaces in humans exhibit a barrier function and are crucial for the defense against pathogens [1]. In addition, they are home to microbiota [2], which, in turn, are part of the epithelial barrier function [1,3]. The study of microbiota has sparked scientific interest in recent years, due to the presumed connection to the general well-being of the body and maintaining local and systemic homeostasis [4,5,6]. Examples of specific, local microbiota are the oral [7], respiratory [8,9,10], skin [11], urogenital [12,13,14], vaginal [15,16], or gastro-intestinal microbiota [4]. When undisturbed and healthy, there is usually a well-balanced and beneficial symbiosis between the microbiota and their host. While the host provides habitat and nutrients for the microbiota, they, in turn, contribute to host homeostasis, since they prevent colonization by pathogens and interact with the innate and adaptive immune system [10,17,18,19,20].

Due to this symbiosis between microbiota and their host, alterations in the microbiota may have potentially beneficial, as well as detrimental, effects [21,22,23,24]. Dysbiosis describes a shift in the composition of microbiota, usually towards more harmful than beneficial bacteria. The impaired homeostasis increases the susceptibility for infection and inflammation. This connection has been shown for the oral microbiota and periodontal diseases and caries [25], gastrointestinal microbiota and various gastrointestinal disorders [20,26,27,28,29], respiratory microbiota of the upper respiratory tract and infections thereof [30], and urogenital microbiota and urogenital infections, as well as the formation of kidney stones [16,31,32].

The gut microbiome, although locally confined, can affect the entire body via gut–brain signaling [33]. The gut microbiome plays an important role, regarding proper functioning of the immune system [34], and changes of the gut microbiome may promote the manifestation of allergies [35] and auto-immune diseases [36]. Moreover, the gut microbiome can influence brain development [37], and changes of the gut microbiome are associated with the occurrence of disorders, such as depression [38], Alzheimer’s disease [39], and schizophrenia [40,41]. A disturbed gut microbiome may contribute to chronic, low-grade systemic inflammation, thus even promoting age-related diseases (inflammaging) [2,37,42,43]. Thus, the gut microbiome is considered one of the most important microbiota that can impact the entire body and its physiology.

## 2. Antibiotic Treatment of Infections: Inappropriate Use and Risks

Although antibiotics are known to be beneficial for the treatment of bacterial infections only, they are often prematurely prescribed, regardless of whether an infection is of bacterial or viral origin [44,45]. The inappropriate use of antibiotics (e.g., for a viral infection) will provide no great beneficial effect, but can be harmful to the patient, as antibiotics impact the microbiota, as well [46]. Even in the case of bacterial infections, for which, in principle, antibiotics are the appropriate treatment, antibiotic treatment is often not required, as many bacterial infections are self-limiting and resolve without treatment. The use of antibiotics can, however, put the patient at risk of adverse effects. For instance, antibiotic use can increase the risk of developing vaginal candidiasis [47].

Further, antibiotics increase the risk for the development of antimicrobial resistance [48,49,50], which globally caused an estimated 1.27 million deaths in 2019 [51]. This is particularly concerning, as there is still a considerable overuse of antibiotics, as an analysis of German outpatient care revealed [52]. Likewise, antibiotics consumption is increasing on a global scale, with low- and middle-income countries converging to levels typically observed in high-income countries. Additionally, a worldwide increase in last-resort compounds has been noted [53]. While the European Centre for Disease Control and Prevention found a decrease in the use of some antibiotics, it, nevertheless, noted an increase in various broad-spectrum antibiotics in the community and hospital sectors [54]. Especially in infants and young children, the overuse and misuse of antibiotics and subsequent effects on the microbiota may contribute to the manifestations of diseases later in life [55,56]. Factors for the irrational use of antibiotics are lack of public knowledge and awareness, access to antibiotics without prescription and leftover antibiotics, pharmaceutical promotion, and inadequate medical training, among others [57]. Ultimately, antibiotics jeopardize stable microbiota, which can have a negative impact on health that may be longer lasting than the often self-limiting and uncritical infections they were initially prescribed for.

## 3. Common Treatment Alternatives to Antibiotics

Antibiotic stewardship, i.e., promoting the responsible and efficient use of antibiotics, is becoming increasingly more common; it has also been addressed in guidelines, such as that from the European Association of Urology (EAU) [58] or World Health Organization (WHO) [59], as well as in various consensus papers and reviews [60,61]. The Centers for Disease Control and Prevention (CDC) define the goals of antibiotic stewardship as “to improve antibiotic prescribing by clinicians and use by patients so that antibiotics are only prescribed and used when needed; … to ensure that the right drug, dose, and duration are selected when an antibiotic is needed.” (https://www.cdc.gov/oralhealth/infectioncontrol/faqs/antibiotic-stewardship.html, accessed on 7 July 2022).

Accordingly, nowadays, antibiotic treatment is rarely recommended for uncomplicated respiratory infections, which are often of viral origin [62,63]. In contrast, antibiotic treatment is often still used in clinical routine for other common infections, such as urinary tract infections (UTI) [64,65]. Yet, for uncomplicated cases of UTI, symptomatic treatment with non-steroidal anti-inflammatory drugs (NSAIDs) is considered a viable treatment alternative to antibiotics [66,67,68,69]. As with any drug, NSAIDs can have adverse effects [70]; although they target inflammation, not bacteria, NSAIDs can impact the gut microbiota and, in turn, negatively affect the outcome of NSAID-therapy itself [71,72].

In addition, many minor and uncomplicated infections are self-limiting and do only require symptomatic treatment. Supporting the natural recovery, by e.g., resting, proper hydration, and avoidance of potential stressors (e.g., alcohol or nicotine), may often be sufficient to overcome uncomplicated infections [73]. However, safer medical treatments, compared to antibiotics or NSAIDs, are still desirable, in order to relieve symptoms and improve quality of life.

## 4. Herbal Drugs: A Safe Treatment Alternative for Uncomplicated Infections

Medicinal products based on herbal drugs or extracts thereof generally exhibit a positive benefit-risk-ratio and are a viable treatment alternative for uncomplicated infections [74]. Unlike antibiotics or NSAIDs, herbal treatment options usually do not target specific pathogens or signaling pathways. Rather, their efficacy is based on a multi-targeted approach [75,76,77]. For many common and recurring infections, such as urogenital infections [78,79] or infections of the upper and lower respiratory tract [80,81], effective and safe herbal treatment options are available. For instance, herbal treatment options for uncomplicated UTI include *Centaurii herba*, *Levistici radix*, and *Rosmarini folium* (Canephron^®^, Bio-norica SE, Neumarkt in der Oberpfalz, Germany) [82], *Tropaeoli herba* and *Armoraciae radix* (Angocin^®^, Repha GmbH, Langenhagen, Germany) [83,84], *Ononidis radix, Orthosiphonis folium,* and *Solidaginis herba* (Aqualibra^®^, MEDICE Arzneimittel Pütter GmbH & Co. KG, Iserlohn, Germany) [85], *Arctostaphylos uva-ursi* (e.g., Cystinol^®^, Schaper & Brümmer GmbH & Co. KG, Salzgitter, Germany) [86], and cranberry [87]. Likewise, for rhinosinusitis, various herbal medicinal products exist, such as *Sambuci flos*, *Gentianae radix*, *Primulae flos*, *Rumicis herba*, and *Verbenae herba* (Sinupret^®^, Bionorica SE, Germany) [88,89], cineole (e.g., Soledum^®^, Cassella-med GmbH & Co. KG, Köln, Germany) [90], myrtol (Gelomyrtol^®^, G. Pohl-Boskamp GmbH & Co. KG, Hohenlockstedt, Germany) [91], and *Pelargonium sidoides* (Umckaloabo^®^, Dr. Willmar Schwabe GmbH & Co. KG, Karlsruhe, Germany) [92,93,94]. Herbal treatment options for acute bronchitis/acute cough include *Thymi herba* and *Primulae radix* or *Thymi herba* and *Hederae folium* (Bronchipret^®^, Bionorica SE, Germany) [95,96], *Pelargonium sidoides* [92,97,98,99], cineole [100], myrtol [101,102], and *Hederae folium* monopreparations (Prospan^®^, Engelhard Arzneimittel GmbH & Co. KG, Niederdorfelden, Germany) [103,104,105]. Importantly, the studies with these products demonstrate that herbal treatment can be effective in reducing symptoms, and, thereby, patient use of antibiotics, while providing generally favorable safety profiles. Moreover, these studies have led to the acknowledgement of herbal medicinal products in guidelines for rhinosinusitis, acute and chronic cough [60,106,107,108], and urinary infections as viable and adequate therapy options [58,109]. In addition, an independent institute (Institute for Quality and Efficiency in Health Care), (IQWiG) attests to Canephron^®^ as having a beneficial effect in cases of recurrent cystitis [110].

## 5. Biologically Active Compounds of Herbal Preparations

By nature, herbal medicinal products are multicomponent mixtures and contain many, often unidentified, active substances. However, some constituents with relevant pharmacological activity have been identified. For example, in vitro studies demonstrated the antiviral activity of various constituents of medicinal plants, e.g., quercetin, carvacrol, or theaflavins [111], as well as antibacterial activity of flavonoids [112], isothiocyanides [113], hydroquinone, and umbelliferone [114].

Anti-inflammatory activity of flavonoids, such as apigenin, quercetin, and kaempferol, as well as that of a variety of other plant constituents, e.g., ursolic acid, betulinic acid, and resveratrol, have been described extensively [115,116,117].

Plants that contain these or other compounds with antiviral, antibacterial, or anti-inflammatory activity promise to be effective treatment options for common and uncomplicated infectious diseases.

## 6. Treatment of Respiratory Infections with Herbal Medicinal Products: Bronchipret^®^ and Sinupret^®^

Infections of the upper and lower respiratory tract (i.e., (rhino-)sinusitis and bronchitis), are considered some of the most common and widespread infections. While antibiotic treatment is common, respiratory infections also respond well to treatment with herbal combinations. A recent review affirmed the growing body of evidence for the effectiveness of herbal products as a treatment of acute rhinosinusitis [118], which is in line with findings from a real-world study that discussed herbal products as a viable alternative to antibiotics [119]. In addition, guidelines for acute and chronic cough suggest herbal products for uncomplicated respiratory infections, in combination with delayed prescription of antibiotics, i.e., patients could receive prescriptions for antibiotics with no further consultation, in case an infection persists, ultimately resulting in notably fewer patients who will take antibiotics, compared to a prescription at the first consultation [108]. Avoiding antibiotics, thus preserving the microbiota, is in line with antibiotic stewardship and beneficial for patients, since the microbiota play a protective role in host defense against respiratory infections [120]. This topic is also of particular importance with regard to children, for whom the effects of overuse and misuse of antibiotics can contribute to the manifestations of diseases later in life [55,56]. Respiratory infections are particularly common in infants and children, with children often suffering from multiple episodes per year [121]. Simultaneously, respiratory infections are associated with frequent medical consultations and an overuse of antibiotics [122]. Moreover, antibiotic use and the frequency of visits are correlated [123]. There is evidence that programs on communication strategies and antibiotic prescribing are successful in decreasing visits [124]. However, these data also emphasize the need for safer alternatives for children, such as herbal treatment.

Two examples of herbal combinations are thyme and ivy or thyme and primrose (as in Bronchipret^®^, Bionorica SE, Germany) for acute bronchitis [95,96] or cowslip, yellow gentian, black elder, common sorrel, and vervain (as in Sinupret^®^ extract, Bionorica SE, Germany) for paranasal sinus infections/rhinosinusitis [88,89].

The efficacy of the thyme-based product Bronchipret^®^ was demonstrated in two prospective double-blind, placebo-controlled clinical trials: one with 361 patients who received an 11-day treatment with Bronchipret^®^ syrup (Bionorica SE, Germany, 5.4 mL, 3 times a day, *n* = 182) or placebo (*n* = 179), and one with 361 patients who received an 11-day treatment with Bronchipret^®^ film-coated tablets (Bionorica SE, Germany, 1 tablet, 3 times a day, *n* = 183) or placebo (*n* = 178). Both trials showed a significantly faster reduction of coughing fits, in comparison to the placebo, as well as a faster regression of bronchitis-related symptoms and higher responder rates with the herbal product [95,96]. In addition to anti-inflammatory and -viral [125] effects, the thyme-ivy/thyme-primula combinations also showed mucus-regulatory effects in acute and chronic bronchitis and bronchoalveolitis [126,127,128].

Similar effects were observed for the respective herbal combinations (Sinupret^®^, Bio-norica SE, Germany) for the treatment of (rhino-)sinusitis, which also exhibited anti-inflammatory and -viral [129] effects and improved mucociliary clearance [130,131,132,133,134]. The efficacy and safety of the herbal medicinal product Sinupret^®^ extract (Bionorica SE, Germany) was shown in a double-blind, randomized, placebo-controlled trial with 386 patients who received either Sinupret^®^ extract (Bionorica SE, Germany, 1 tablet 3 times a day) or matched placebo: the treatment resulted in significant, clinically relevant differences in the major symptom score (MSS), in favor of the herbal product, thus leading to two days earlier symptom relief, better quality of life, and higher responder rates, compared to the placebo [88].

## 7. Preservation of the Gut Microbiome under BNO 2811 and BNO 1011: Results of a Mouse Model

While herbal medical products are assumed to preserve the gut microbiome, the impact on the microbiome has, to date, not been well-studied for respiratory infections. In the following, we present some initial, thus far unpublished, preclinical data for BNO 2811 (mixture of ethanolic dry extract of *Thymi herba* and dry extract of *Primulae radix*) and BNO 1011 (ethanolic dry extract of a mixture of *Gentianae radix*, *Primulae flos, Rumicis herba, Sambuci flos,* and *Verbenae herba*), which are the basis for Bronchipret^®^ film-coated tablets and Sinupret^®^ extract (Bionorica SE, Germany).

To analyze the impact of these herbal combinations and first-line antibiotics for the treatment of respiratory infections on the gut microbiome, compositions of the fecal microbiome from mice were analyzed via next-generation sequencing (NGS) of bacterial 16S rRNA genes using a quality-controlled workflow [135]. The mice received either daily oral doses of the antibiotics amoxicillin/ clavulanic acid or moxifloxacin or the herbal extracts BNO 2811 (one-fold equivalent of the recommended daily human dose of Bronchipret^®^ film-coated tablets, Bionorica SE, Germany) or BNO 1011 (one-fold equivalent of the recommended daily human dose of Sinupret^®^ extract, Bionorica SE, Germany). An additional group was fed with water, which served as a substance-free vehicle/control group. Fecal samples from four animals per treatment arm were taken after seven days of treatment (Figure 1).

NGS-based analyses of the microbiome revealed a significant alteration of the bacterial composition during antibiotic treatment, while the microbiome of mice that had been fed with herbal extracts was very similar to substance-free vehicle controls (Figure 1A,B). The most significant impact was observed after the gavage of amoxicillin/clavulanic acid, which led to a marked loss of bacterial diversity, accompanied with the domination of only few genera (*Enterobacteriaceae* species, *Escherichia-Shigella, Parabacteroids, Robinsoniella*). To further assess the long-term effects on the intestinal microbiome, the treatment of mice with amoxicillin/clavulanic acid was discontinued, and the fecal microbiome was again analyzed after an additional 11 weeks (d84). Microbial compositions again changed, but did not return to baseline after this prolonged period. In addition, potentially beneficial species, such as *Akkermansia muciniphila*, did not reappear after treatment termination. Thus, antibiotic treatment led to long-lasting changes of the bacterial microbiome.

## 8. Treatment of Urogenital Infections with an Herbal Medicinal Product: Canephron^®^

To illustrate an effective and safe herbal treatment option for uncomplicated UTIs, the herbal medicinal product Canephron^®^ (Bionorica SE, Germany), which contains the phytocombination BNO 2103 of *Rosmarini folium*, *C**entaurii herba*, and *L**evistici radix* as active pharmaceutical ingredient, is discussed.

The efficacy of Canephron^®^ N (BNO 1045, Bionorica SE, Germany) has been shown in a double-blind, placebo-controlled, randomized clinical trial [82]. In this trial, 325 women were randomized to treatment with BNO 1045, and 334 women were randomized to antibiotic treatment with fosfomycin trometamol. The results demonstrate the non-inferiority of BNO 1045 versus antibiotic treatment in acute lower uncomplicated UTI, with regard to the need of additional antibiotic treatment.

In addition to efficacy, the effectiveness under real-world conditions has been shown in a recently published study based on a real-world database analysis reviewing over 160,000 cases of UTI treatment with either antibiotics or Canephron^®^ (Bionorica SE, Germany) [136]. The findings of this study confirm the results of the above-mentioned clinical trial [82]. The percentage of patients needing additional antibiotics from day 1 to 30 was almost identical for both groups. In addition, the probability of sporadic or frequent recurrent UTI episodes was lower following treatment with the herbal medicinal product. Surprisingly, the need for additional antibiotic treatment from day 31 to 356 was higher in the group of patients who received antibiotics as an initial treatment of uncomplicated UTI [136].

A potential explanation for this may be the altering impact of antibiotics on specific microbiota. Healthy gut, vaginal, and urinary microbiota are thought to protect from urinary infections; accordingly, dysbiosis is implicated in the etiology of UTIs [137]. Herbal medicinal products, on the other hand, are thought to preserve the microbiota and, thus, its protective role. The therapeutic effect of herbal treatments, which are multi-component mixtures with typically more than one mode of action, can be explained by therapeutic effects other than antibiotic.

## 9. Preservation of the Gut Microbiome under BNO 2103: Results in a Mouse Model

The herbal combination BNO 2103 is known to be efficacious for treating urogenital infections in humans by impeding the adhesion of pathogens in the urogenital tract, as well as having spasmolytic, diuretic, anti-oxidative, anti-inflammatory, and anti-nociceptive effects that contribute to the successful treatment [82,136,138,139,140,141,142,143]. Further, in contrast to antibiotics and NSAIDs, herbal medical products are thought to preserve the gut microbiome. However, the impact of the treatment on the gut microbiome has not yet been thoroughly investigated preclinically.

In 2017, Naber and colleagues published preliminary findings on the effects of this herbal combination on the gut microbiome in mice [144]. To further test the impact of the treatment on the gut microbiome, stool samples from mice were analyzed by next-generation sequencing of bacterial 16S rRNA genes using a quality-controlled workflow [135]. The mice received daily oral doses for 7 days of the antibiotic nitrofurantoin, water (as a substance-free vehicle; control group), phytocombination BNO 2103, or a single dose of the antibiotic fosfomycin on day 1. The dosages of BNO 2103 were 65 and 1333 mg/kg, which is equivalent to one- and twenty-fold the recommended human dosage of Cane-phron^®^ (Bionorica SE, Germany). Each arm of the study comprised four animals, and stool samples collected prior to treatment, on day 2 (for fosfomycin-treated mice) or 7 (remaining groups) were analyzed. All mice were handled, and the experiments were conducted, with the approval of, and in compliance with, the institutional guidelines and respective authorities (District Government of Lower Franconia).

The sequencing results revealed considerable shifts in the composition of the gut microbiome under treatment with nitrofurantoin. The changes were more distinct in the fosfomycin-arm of the experiment: with just a single dose, some bacterial families had completely disappeared from the gut microbiome, and they had not recovered during the following days without treatment. The phytotherapeutically-treated mice displayed a mostly unaltered diversity of gut bacteria, similar to that of the control group of mice receiving (substance-free) water. Even when receiving the 20-fold equivalent of the recommended human dosage, the gut microbiome of the mice was hardly altered (Figure 2A,B). These findings support the microbiota-sparing effects of BNO 2103 and contribute to the existing body of evidence regarding the favorable safety profile of the phytocombination.

## 10. Future and Prospects for Application

The overuse and misuse of antibiotics remains a challenge. Antibiotics can induce harmful shifts in the microbiota, with consequent negative effects on health that may last longer or be more severe than the initially treated infection itself [46]. When facing self-limiting and uncomplicated infections, antibiotics can be considered an overtreatment; when used for viral infections, they are entirely inappropriate [44,45]. Especially in infants and young children, the inappropriate use of antibiotics can be detrimental, as the subsequent effects of antibiotics on the microbiota can contribute to the manifestation of diseases later in life [55,56]. Additionally, for minor infections, the frequent medical consultation that is also associated with higher antibiotic use can divert resources from the care of potentially more serious conditions [122,123].

A further problem of the overuse and misuse of antibiotics is the increased risk for the development of antimicrobial resistance [48,49,50], which caused an estimated 1.27 million deaths worldwide in 2019 [51].

While awareness of antibiotic stewardship is growing, generating a more widespread understanding for the responsible use of antibiotics remains important, in order to reduce the risk of adverse effects and antibiotic resistances, but also to promote a more conscious treatment choice for infections [145].

Treatment alternatives, such as NSAIDs, do not contribute to antibiotic resistance, but can still impact the microbiota. This, in turn, may introduce other impairments, despite resolving symptoms of the initial infection. As our understanding of the microbiota and its association to the general well-being and resilience towards diseases has increased, it has become apparent that the preservation of the microbiota must be considered when choosing an appropriate therapy for infections.

It has been demonstrated that relapse rates were lower with a phytocombination than with antibiotics in UTI [136]; by stabilizing the urogenital and intestinal microbiota, the natural immune response can, ultimately, also be strengthened [146]. Herbal extracts can be alternatives to antibiotics and NSAIDs for the treatment of uncomplicated urogenital and respiratory infections. Importantly, data show that herbal medicinal products can provide a comparable efficacy to antibiotic and NSAID treatment for UTIs and offer a generally favorable safety profile [82,95,96]. One reason for this may be that herbal treatments do not impact the gut microbiome, as shown in a mouse model for BNO 1011, 2811, and 2103 [135]. However, herbal medicinal products cannot replace antibiotics in all instances; in cases of uncomplicated infections, delayed prescription of antibiotics, in favor of starting treatment with herbal medicinal products, may be useful for reducing the use of antibiotics [147,148]. It is crucial to increase public knowledge and awareness, as well as provide appropriate medical training and communication strategies, in order to prevent overuse and misuse of antibiotics, especially when alternatives are available [57,124]. Overall, this review aims to emphasize the contribution of herbal preparations to antibiotic stewardship with low risk of negative impact on patients’ microbiota and well-being.

## Figures and Tables

**Figure 1 antibiotics-11-01331-f001:**
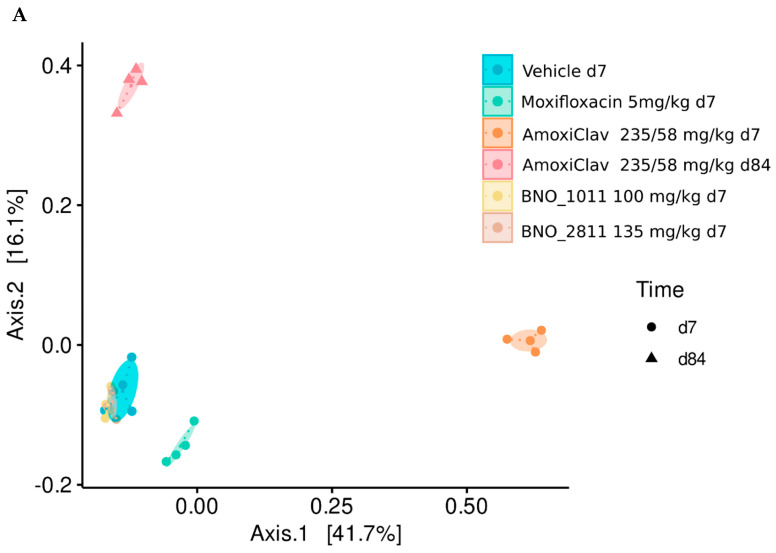
Changes in microbiome after treatment of mice with amoxicillin/clavulanic acid (day 7 and 84), moxifloxacin (day 7), BNO 2811 (day 7), or BNO 1011 (day 7), compared to a control group (water). (**A**) Similarity between individual bacterial compositions were studied using principal coordinates analysis (PCoA) of 16S rRNA gene sequencing data. Individual samples (colored dots) clustered well, according to the treatment groups. Ellipses represent the 95% confidence intervals, based on a multivariate t-distribution for each group. The center of each group is marked by small dots. Differential clustering of treatment groups after PCoA indicates compositional shifts after seven days of antibiotic treatment with amoxicillin/clavulanic acid (orange dots). Additional shifts of bacterial compositions 11 weeks after discontinuation of amoxicillin/clavulanic acid (d84, red triangles) point to a long-term damage of the microbiome, due to the antibiotic treatment. Bacterial compositions of mice treated with BNO 1011 and BNO 2811 showed high similarity to untreated mice, inferring no impact on the intestinal microbiome. Coordinates represent 41.4 and 16.1 percent variance of the dataset. (**B**) Taxonomy bar plot illustrating relative abundances of detected bacterial genera in samples and treatment groups.

**Figure 2 antibiotics-11-01331-f002:**
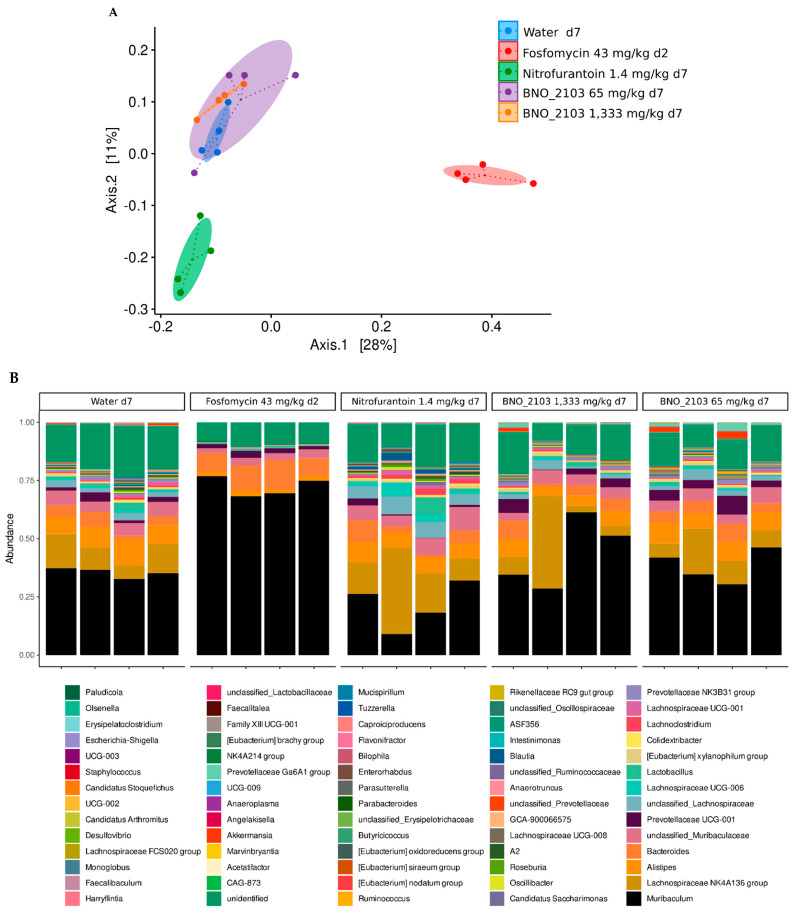
Changes in microbiome after treatment of mice with fosfomycin (day 2 after a single dose), nitrofurantoin (day 7), and BNO 2103 (day 7), compared to a control group (water, day 7). (**A**) Similarity between individual bacterial compositions were studied using principal coordinates analysis (PCoA) of 16S rRNA gene sequencing data. Individual samples (colored dots) clustered well, according to the treatment groups. Ellipses represent the 95% confidence intervals, based on a multivariate t-distribution for each group. The center of each group is marked by small dots. Differential clustering of samples after treatment with fosfomycin or nitrofurantoin indicates compositional shifts after antibiotic treatment of mice, while mice treated with BNO 2103 or controls are clustering together, denoting similar bacterial compositions. Coordinates represent 28 and 11 percent of total variance of the dataset. (**B**) Taxonomy bar plot illustrating relative abundance of detected bacterial genera in samples and treatment groups.

## Data Availability

The data presented in this study are available upon request from the corresponding author. The data are not publicly available, due to data ownership by Bionorica SE.

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
