# Peer review of "Contribution of Symptomatic, Herbal Treatment Options to Antibiotic Stewardship and Microbiotic Health"

_antibiotics, 2022, doi:10.3390/antibiotics11101331_

Round 1
Reviewer 1 Report
The article presents different herbal treatments that could be used as an alternative in the management of common, uncomplicated infections, in order to preserve the health of the microbiota. I found the article to be quite interesting, well organized and described, even though the authors mainly reffered to just a few herbal preparations which are frequently administered.
Overall, this is a review that aims to emphasize the contribution of herbal preparations to antibiotic stewardship with low risk of negative impact on patients’ microbiota and well-being. The topic is interesting and relevant in the field, although it is not necessarily original. Recently, there have been an increasing number of studies which have investigated the interaction between the gut microbiota and herbal medicines. The methodology is appropriate. Unpublished preclinical data regarding a mixture of ethanolic dry extract of Thymi herba and dry extract of Primulae radix, as well as regarding an ethanolic dry extract of a mixture of Gentianae radix, Primulae flos, Rumicis herba, Sambuci flos, and Verbenae herba are presented. The conclusions are consistent with the evidence presented in the article and they address the main question posed. The references are appropriate and comprehensive.
Minor comment
Line 230: it should be "a herbal medicinal product, instead of "an". Otherwise, the language is professional.
Author Response
Dear Reviewer
Thank you very much for your thorough, thoughtful, and prompt review of our manuscript “Contribution of symptomatic, herbal treatment options to antibiotic stewardship and microbiotic health” (antibiotics 1903073) and for recommendations for its improvement.
We implemented all of your corrections/recommendations. In addition, some mistakes/typos have been corrected.
Please find attached the revised manuscript with all changes marked up using the “Track Changes” function or by comments for further review.
Thank you and kindest regards,
André Gessner

Reviewer 2 Report
The authors submitted the good manuscript "Contribution of symptomatic, herbal treatment options to antibiotic stewardship and microbiotic health", but it should be better structured. Due to a large amount of information, it is difficult to read it in its current form. The manuscript should be divided into subsections, for example, "introduction", "groups of antibiotics and their use", "herbal preparations (further can be divided into subgroups: for the treatment of intestinal diseases, respiratory diseases, etc.", "use in pediatrics", " side effects of herbal preparations».
I recommend that the authors add a subsection "Biologically active compounds of herbal preparations" that can be used to treat simple infectious diseases, where a few of the most active ingredients of plant origin are described.
In addition, you can add some information on the marketing analysis of herbal preparations (which countries produce, how many components, dosage forms, etc.). Specify which and how many brands and generics. This is at the discretion of the authors.
Conclusions should be changed to “future and prospects for application”. Thus, the article requires minor clarifications and additions.
Author Response
Dear Reviewer
Thank you very much for your thorough, thoughtful, and prompt review of our manuscript “Contribution of symptomatic, herbal treatment options to antibiotic stewardship and microbiotic health” (antibiotics 1903073) and for recommendations for its improvement.
We implemented all corrections/recommendations, except the addition of information on the marketing analysis of herbal preparations, because this is, in our opinion, beyond the scope of the manuscript, which already contains a large amount of information.
In addition, some mistakes/typos have been corrected.
Please find attached the revised manuscript with all changes marked up using the “Track Changes” function or by comments for further review.
Thank you and kindest regards,
André Gessner

Reviewer 3 Report
First of all, I would like to congratulate the authors of the review, as they have done an excellent job.
The review perfectly meets the scope of Antibiotics and deals with a topic of great interest today.
The text and information it contains is very well reviewed and can hardly be faulted. As for specific changes to the text, I propose the following:
- Change the format of the headings of the sections to make them more visible and differentiate them from the body of the text.
- Line 257: abbreviate the names of Rosmarini folium, Centaurii herba, and Levistici radix, as they have appeared earlier in the text.
- Line 333 and others: remove the "e.g." inside the brackets of the references. I think it's unnecessary and it could be better without it.
- Refs 118 and 135: remove the DOI link to standardize the reference format.
Author Response

(The authors gave the same response as above.)

Round 2
Reviewer 2 Report
Good scientific article, it deserves to be published.